



# Elevation Change of the Greenland Ice Sheet and its Peripheral Glaciers: 1992-2023

Johan Nilsson[1] and Alex S. Gardner[1]

[1]Jet Propulsion Laboratory, California Institute of Technology, Pasadena, 91109, United States of America

*Correspondence to*: Johan Nilsson (johan.nilsson@jpl.nasa.gov)

**Abstract.**

The integration of data from multiple satellite altimetry missions, each offering unique observational characteristics, has enabled us to discern both short-term variability and long-term climate trends affecting Greenland's peripheral glaciers and the Greenland Ice Sheet (GrIS). Our methodology, adapted and informed by lessons learned from our similar efforts in

Antarctica, ensures the consistency and reliability of the derived elevation change dataset. The data covers the years 1992-2023 and is made publicly available as part of the NASA's Making Earth System Data Records for Use in Research Environments (MEaSUREs) Inter-Mission Time Series of Land Ice Velocity and Elevation (ITS_LIVE) project. Our analysis of the dataset reveals significant patterns of mass loss across the GrIS. We find that the ice sheet and peripheral glaciers have experienced an average mass loss of -173 $\pm$ 19 Gt a$^{-1}$ and -23 $\pm$ 5 Gt a$^{-1}$ respectively over the 1992-2022 time period (given

temporal availability of selected firn models), with notable variations over time. Specifically, the early years of the record exhibit a positive mass balance, likely due to anomalously positive surface mass balance. However, this trend shifts in subsequent years, with a pronounced increase in mass loss rates, highlighting the accelerating impact of a changing climate on ice sheet dynamics and surface mass balance. Moreover, our analysis underscores the importance of considering peripheral glaciers in addition to the continental ice sheet when assessing overall mass trends. By incorporating data from peripheral

glaciers, we provide a more comprehensive understanding of Greenland's total contributions to global sea level rise. Our findings reveal not only the magnitude of mass loss but also its evolution over time, emphasizing the need for continued monitoring and research to better understand the impacts of climate change on Earth's cryosphere.

## 1 Introduction

The Greenland Ice Sheet (GrIS) is the second largest reservoir of freshwater, after the Antarctic Ice Sheet (AIS), and has the

potential to raise global sea levels by approximately seven meters if completely melted (Houghton, 2004). Currently, it is the one of the largest contributor to eustatic sea level rise, surpassing the Antarctic Ice Sheet (Otosaka et al., 2023) and comparable to the rest of the world's glaciers. It is projected that by the end of the century, the GrIS will contribute over 30 cm to global sea levels under high-emission scenarios (Hugonnet et al., 2021). Over the past three decades, starting from the onset of the satellite altimetry era in 1992, it is estimated that the GrIS lost nearly 3.8 trillion tons of ice, resulting in a global sea level rise

of approximately 11 mm (Shepherd et al., 2018). Satellite altimetry plays a crucial role in studying changes in the cryosphere at continental scale. The remoteness and vast expanse of the world's ice sheets necessitate long-term space-based remote sensing to distinguish short-term variability from long-term climate trends. The satellite altimetry record for cryosphere studies began in the early 1992 and continues to the present day, incorporating several altimetry missions with varying characteristics (e.g., radar and laser altimeters).


Generating continuous time series of elevation change from satellite altimetry is a complex task with several challenging processing steps. These steps include geophysical corrections, waveform retracking, static topography removal, scattering regime corrections, cross-calibration, and more. In this study, we present a new dataset that incorporates all of the aforementioned corrections, enabling the generation of a continuous elevation change dataset for Greenland that spans the

period from 1992 to 2023. Using the methodology developed in an earlier study of Antarctic Ice Sheet elevation change ( Nilsson et al. 2022), we synthesize data from six different satellite altimetry missions collected over Greenland, comprising of both radar and laser altimeters. The resulting dataset has undergone comprehensive validation using independent, unbiased airborne laser altimetry data. This effort is part of the ITS_LIVE project (https://its-live.jpl.nasa.gov/), which has been developed over the last few years to generate synthesized and continuous elevation, velocity and ice extent records for both

the Greenland and Antarctic Ice Sheets. ITS_LIVE has already published a longer record of elevation change for the Antarctic Ice Sheet, covering the period from 1985 to 2020. The lessons learned and methodology from the processing of the Antarctic data have been applied to this new product for Greenland.

This study represents the most complete synthesis of satellite altimetry data to date, covering the period from 1992 to 2023.

Notably, it includes not only the continental ice sheet but also three decades of change of Greenland's peripheral glaciers for the first time. This record was used in combination with two firn models to estimate that Greenland (continental ice sheet plus periphery glaciers) lost a total of $6076 \pm 609$ Gt of ice over the 1992-2022 (given temporal coverage of selected firn models) time period with large inter-decadal variability. The ice sheet and peripheral glaciers had an mass budget of $+46 \pm 76$ Gt a$^{-1}$ in the first decade of the record, a budget of $-362 \pm 53$ Gt a$^{-1}$ in the second decade, and a rate of $-144 \pm 39$ Gt a$^{-1}$ in that last

decade.

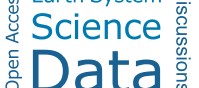

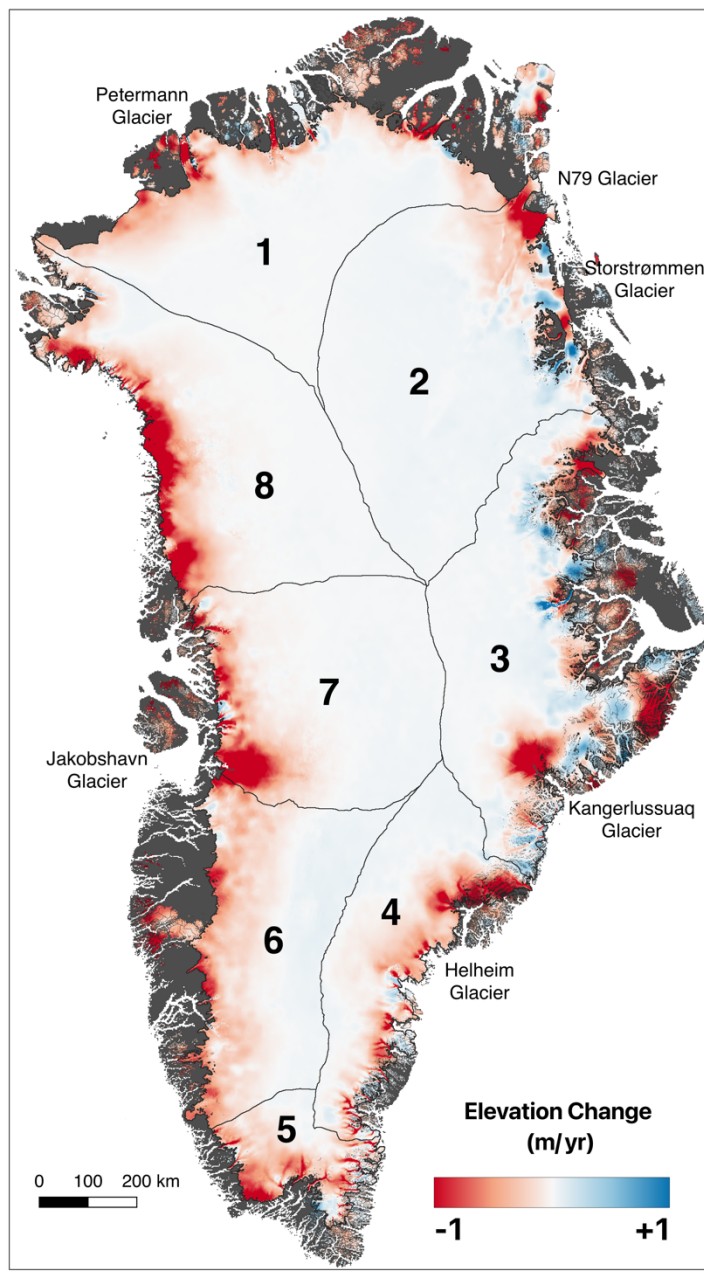

**Figure 1: Elevation change of the Greenland Ice Sheet and glacier from 1992 to 2023. Drainage basins 1–8 (Zwally et al., 2012) shown in black with labelling of large outlet glaciers.**




**2 Data and coverage**

In this study we utilized a total of six missions to obtain a 32-year long time series of elevation changes for the GrIS and its peripheral glaciers spanning the period from 1992 to 2023. The missions encompassed both radar and laser altimeters that operated during different overlapping time intervals: the European Remote Sensing (ERS) satellites ERS-1 (1992-1996) and
ERS-2 (1995-2003), the Environmental Satellite (Envisat) (2003-2012), the Ice, Cloud, and land Elevation Satellites (ICESat) (2003-2009) and ICESat-2 (2018-Present), and CryoSat-2 (2010-Present). The methodology for geophysical corrections and data editing for each mission follows the same approach as described in the reference paper of Nilsson et al. (2022), with a few minor modifications that we detail here. For Envisat, we utilized the entire time period of operation. This differs from Nilsson et al. (2022), which only used Envisat data through to the orbit change that occurred in late October of 2010. Including
the additional 17 months of data enabled better overlap with CryoSat-2. Additionally, we employed the latest fully reprocessed Envisat data, specifically the "RA-2 Level 2 products: the Geophysical Data Record (GDR) version 3.0" (Anon, 2018). For CryoSat-2 we use the updated L1b Baseline-E for both the Low-Resolution Mode (LRM) and the Interferometric Synthetic Aperture Radar (SARIn) modes (available at science-pds.cryosat.esa.int - last checked 2024-06-11). For ICESat we apply Laser-2/Laser-3 biases based on the recommendation provided in Borsa et al. (2019), this was not done in Nilsson et al. (2022).
For ICESat-2 we utilized release-006 of the ATL06 (Smith et al., 2023). Otherwise we use the same data and follow all of the processing steps described in Nilsson et al. (2022) that we summarize in the Methods.

To convert from volume change to mass change we utilized two different Firn Densification Models (FDM) to account for changes in the Firn Air Content (FAC) that result in changes in ice volume without any corresponding change in mass. We use
the Glacier Energy and Mass Balance FDM (GEMB) version 1.2 (Gardner et al., 2023) and the Goddard Space Flight Center FDM version 1.2.1 (GSFC) (Medley et al., 2022). Both products contain model results for the continental ice sheet and peripheral glaciers. The modelled FAC was interpolated in space and time from their native output to our 1920 m resolution and monthly temporal sampling using simple linear interpolation through the end of 2022 (temporal coverage of both model). To compute mass change, FAC volume changes were subtracted from glacier volume changes to determine the ice equivalent
volume change. The ice equivalent volume change was then multiplied by the density of ice ($917 \, \text{kg m}^{-3}$) to determine mass change from the two models, and the final mass change estimate is obtained by taking the mean of two estimates. To delineate the continental ice sheet we use a data set provided by F. Paul (Bolch et al., 2013) and for its drainage basins we used outlines from Zwally et al. (2012) . For  peripheral glaciers, we used Region 5 of the "Randolph Glacier Inventory 7.0" (RGI) (RGI Consortium, 2023), including only the unconnected glaciers (CL0 and CL1).





## 3 Methods

To derive continuous long-term elevation change estimates for the GrIS we closely follow Nilsson et al. (2022) with some minor updates and improvements.

### 3.1 Reading and editing of altimetry data

Reading and editing of the altimetry data, and application of the standard corrections (dry and wet troposphere correction, ionosphere correction, load tide, pole tide and solid earth tide) followed the same procedure as Nilsson et al. (2022). For ICESat we did not apply an inter-campaign correction following previous Nilsson et al. (2022), but we do apply a 3.1-cm bias between ICESat's Laser 2 and Laser 3 as suggested by Borsa et al. (2019).

### 3.2 Slope correction of pulse-limited radar altimetry data

Over an inclined surface the radar return does not originate from Nadir, but rather from the point-of-closest-approach (POCA). Assuming that the return comes from nadir leads to both incorrect estimate of elevation and the position of the return (Brenner et al., 1983). Due to the large size of the ground footprint of pulse-limited radars (~16 km in diameter – beam-limited) the surface return can originate from anywhere inside this region. To mitigate this error a correction is needed based on the shape of the topography within the pulse-limited footprint. There are several types of corrections available (see Bamber, 1994). In 105 this study we employ the relocation method (Nilsson et al., 2016; Roemer et al., 2007; Schröder et al., 2017) to correct both the position and the elevation of the echo. This was done by using the slope and aspect estimated from the GIMP digital elevation model (Howat et al., 2014) down sampled (by averaging) to 2 km. This correction is not needed for laser altimetry (ICESat and ICESat-2) and for CryoSat-2 SARIn mode because of the small footprint size and the interferometric capabilities, respectively.

### 3.3 Separation of static and time variable topography

To resolve time variable changes in surface elevation the underlying time-averaged topography needs to be removed. Following Nilsson et al. (2022), we separate static and time variable topography by removing a biquadratic surface model from the point data. Time variable height was computed on a 500 x 500 m grid with a 1000 m search radius for CryoSat-2 and a 500 m search radius for all other missions. In contrast to Nilsson et al. (2022), we did not separate ascending and descending tracks 115 when separating separate static and time variable topography. This is because over Greenland we did not detect any ascending/descending biases of consequence, which is in agreement with Schröder et al., (2018).

### 3.4 Scattering correction of radar altimetry data

Most radar altimeters operate in Ku band (13.6 GHz) that is sensitive to radar penetration into snow and ice (Arthern et al., 2001). This sensitivity introduces errors into in the retrieved elevation, causing changes in both the measured trend and the





seasonal variability (Davis and Ferguson, 2004; Khvorostovsky, 2012; Nilsson et al., 2015; Wingham et al., 1998). To mitigate this effect, a local regression is performed between the backscattered power, the leading edge and the trailing edge slope with height change elevation and subtracted from the height data (Flament and Rémy, 2012; Nilsson et al., 2022; Paolo et al., 2016; Simonsen and Sørensen, 2017; Zwally et al., 2005). Here, each dataset is divided into 1x1 km spatial bins and the regression is performed to the data within each bin. Compared to Nilsson et al. (2022) a modification was done to remove the mean for each waveform parameter inside the bin instead of differencing, a similar approach to Flament and Rémy, (2012). Tests around the "The North Greenland Eemian Ice Drilling" (NEEM) station showed improved trends and seasonality when removing the mean, relative to differencing. Hence, care should be taken when selecting this type of operation as it might be dependent on the magnitude of penetration depth and the differences in scattering characteristics for the Greenland and Antarctic ice sheets (e.g. wet versus dry). It was noticed in the previous application of the scattering correction algorithm (Nilsson et al., 2022) in Antarctica that issues could arise when no-data values were present in the individual waveform parameters, providing erroneous solutions. To reduce this issue a simple spatio-temporal nearest neighbour interpolation steps was added to fill no-data values for each waveform parameter before the inversion.

### 3.5 Stacking of multi-mission altimetry data

To create monthly time series of elevation change spatio-temporal binning or "stacking" was employed. Here, we stacked the point-data onto a monthly 1920 x 1920 m regular grid (t,x,y) using a constant search radius of 10-km around each node. This is one of the updates from our previous version of the processing pipeline (Nilsson et al., 2022), which applied the stacking in the cross-calibration step. Here stacking and cross-calibration were separated to provide simplified and efficient re-processing of data. Another improvement in the stacking step was to use an exponential spatio-temporal kernel to generate monthly estimates of elevation change. Here, a 3-month window of data was used to estimate the monthly weighted averages within the 10-km search radius using a correlation length of 2-km that is consistent with the spatial resolution of the grid and the resolution of the pulse limited radar missions. This procedure introduces light interpolation and smoothing to the data, benefiting start and end-points. This also provides improved overlap between missions, which are needed for mission cross-calibration. To reduce noise in monthly estimates, each time series was filtered before stacking by fitting a third order polynomial plus seasonal model to the data and rejecting data falling outside 10-sigma and ±10 m of the residuals. Further, inside each 3-month window we rejected data falling outside 10-sigma. Sigma-editing was performed using the scaled "Median Absolute Deviation" (MAD) as previously used in Nilsson et al. (2022).

### 3.6 Normalization of seasonal amplitudes

It is well documented that radar penetration influences the magnitude of the retrieved seasonal signal observed by radar altimeters (Davis and Ferguson, 2004; Nilsson et al., 2015; Wingham et al., 1998; Khvorostovsky, 2012). The degree and nature of radar penetration is mission dependent. This causes varying observed seasonal amplitudes between missions. A large part of this inter-mission seasonal amplitude variability is corrected for by the scattering correction (Section 3.4) but not all.



This can be observed in the products of Schröder et al. (2018) and Nilsson et al., (2022). To normalize seasonal amplitudes between mission we apply the same method as used in Nilsson et al. (2022). Here, a reference mission is used to create a seasonal correction based on the ratio of between the reference mission and target mission inside each bin. Once this correction
is estimated it is then removed from the target mission aligning its overall seasonal amplitude with the reference. CryoSat-2 was selected as reference mission due to previous experience over Antarctica. The correction was only applied to the radar altimetry (excluding CryoSat-2) data and not to ICESat or ICESat-2. The correction was estimated for each grid node by first removing a 3$^{rd}$ order polynomial and then fitting a seasonal model to each time series. Using the estimated seasonal model, a scaled version based on the amplitude, was created and removed from each radar time series. In the last step the 3$^{rd}$ order
polynomial was added back. See Nilsson et al., (2022) for more details.

### 3.7 Cross-calibration of multi-mission altimetry data

Individual altimetry mission time series need to be cross-calibrated at each grid node to produce an aligned record. This is achieved using an two-step processed outlined in Nilsson et al., (2022) which consists of fitting a polynomial of variable order (order ranging from 1 to 5) with an added annual and semi-annual seasonal model in combination with an overlap bias term.
The model order is determined as in Nilsson et al., (2022) using the "Bayesian information criterion". The coefficients for this model are then solved for using iterative least-squares with a maximum of five iterations in combination with a 10-sigma and ±10 m filter. Due to variable data coverage, mission biases cannot be determined everywhere. We therefore interpolated the bias to each node and remove it from each mission time series. To account for any non-linearities not captured by the selected model, a secondary calibration offset is generated by computing the median difference of the model residuals for each
overlapping time period for each mission. This residual offset after the initial correction is then interpolated and removed in the same manner as before. One should note that to reduce the number of overlaps, the different missions (and modes) are first grouped after the initial least squares adjustment, and the offsets computed on the grouped overlaps. Missions are grouped as follows: ERS-1 (Ocean/Ice-mode), ERS-2 (Ocean/Ice mode), Envisat with ICESat, CryoSat-2 (SARIn/LRM-mode) and ICESat-2 creating a total of five groups (i.e., five overlaps). For Greenland we found that there was no need to provide an
external rate comparison of cross-calibration offsets to the ICESat-2 mission as done in Nilsson et al., (2022). The residual cross-calibration algorithm could then be simplified to only use Solution 1 in Nilsson et al., (2022) where the offset is simply computed by differencing the median values for each overlapping part of the time series for each group.

### 3.8 Interpolation and merging of multi-mission time series

Here, the merging and the interpolation is performed in the same step using the individual measurement error as the weighting
in the interpolation procedure. We update the optimum interpolation routine relative to Nilsson et al., (2022). In Nilsson et al., (2022) the selected background fields, based on velocity and hypsometry, were applied locally using the closest data in horizontal distance. Our analysis showed that the local approach was not robust and that using a background model based on data from a larger region (~150 km) provided improved results. Further modifications to the model were to remove the log-





transform applied to the velocity data. Velocity and hypsometry were binned in 100 m elevation bands at each node using data
within 150 km radius and regressed against the elevation change difference for that specific month.

$$dh = c_0 + c_1v + c_2h_{DEM} \tag{1}$$

were, $v$ is the velocity and $h_{DEM}$ is the hypsometry from the digital elevation model. Ice sheet dynamics are best captured by
changes in surface velocity. However, radar altimetry is unable to measure changes in areas of complex terrain. Studying the
relationship between elevation change and velocity magnitude on a continental wide basis we find a clear linear relationship
up to roughly 300-500 m a$^{-1}$, after which the correlation decreases rapidly. This decrease in correlation is most likely related
to the inability of the altimeter to measure areas of fast flow that are most often located in areas of complex topography, like
outlet glaciers. Because of this, we assume that the linear relationship for low velocities can be extrapolated to areas of higher
velocities. To capture the relationship between velocity magnitude and elevation change in the model we weight the solution
using the inverse squared of the velocity ($w = {}^1/_{v^2}$) in the multivariate regression. No weighting in regards to elevation ($h_{DEM}$)
was done but could be investigate and included in future solutions. For each node we select data within a 150 km radius or the
3000 closest points (if less than 3000 points are available), the coefficients of the model are solved for and subtracted from the
closest 49 points for that node to reduce the computational burden. The residuals of these 49 points compared to the model are
used to predict the value at the grid node using least-squares collocation (optimum interpolation). A second order Gauss-
Markov model is used to model the covariance assuming a 20-km correlation length. To reduce "trackiness" in the gridded
spatial fields a variance of 1 cm$^2$ to the off-diagonal elements of the covariance matrix (similar to Paolo et al., 2023). Once the
value at the grid node has been estimated from the residual data the background model is added back to restore the signal of
each monthly spatial field. This is referred to as the remove and restore technique and has been widely used in geodesy for
decades (Moritz, 1978). Before the interpolation stage a simple spatial filtering is performed. Each monthly spatial field is
divided into 100 km bins and inside each bin a biquadratic surface model is fit to the data. The residuals to this model fit are
used to identify and reject outliers using the "inter-quantile range" (IQR) excluding data below or above the 25th and 75th
percentile respectively inside each bin.

### 3.9 Post filtering and smoothing of product

The final stage of processing involves filtering individual time series to eliminate and replace outliers. This procedure closely
adheres to the methodology outlined by Nilsson et al. (2022). Initially, a LOWESS algorithm (Cleveland, 1979) is employed
to estimate the time series trend, and outliers are identified and removed using a 10-sigma edit based on the median absolute
deviation (MAD) of the residuals with respect to the trend. Subsequently, the residuals are further refined using a moving 12-
month window of an interquartile range (IQR) filter to eliminate any remaining outliers (excluding data below or above the
25th and 75th percentile respectively). Rejected epochs are then filled and slightly smoothed by convolving the time series
with a Gaussian kernel having a correlation length of three months and a filter length of nine points. Any unfilled regions



resulting from these processes are spatially filled using a Gaussian kernel with a correlation length of roughly 20 km for each monthly spatial field, extending over approximately 30 km.

## 4 Error propagation and validation

Airborne laser altimetry has been collected over the GrIS since the early 1990's. More than 88-million-point elevation measurements have been collected by the Airborne Topographic Mapper (ATM) instrument (MacGregor et al., 2021) carried on multiple missions over Greenland starting in 1993 until the end of Operation IceBridge (OIB) in 2019. This makes it a vital source of validation data for space-borne altimeters. We validate our product against ATM data to estimate the uncertainty of our derived surface elevations. For each mission point-to-point elevation difference (altimetry minus airborne) are computed

from data within a search radius of 50 m and a three-month window. The three-month window was selected to obtain enough samples for the older mission. The selection of the temporal window size had little effect on the overall statistics. Statistics are sensitive to the selection of the horizontal search radius and is a trade-off between the number of samples and decreased precision. Our selection of a 50 m search radius is based on previous work (Gray et al., 2017; Nilsson et al., 2016). Elevation differences were then binned by surface slope derived from the GIMP digital elevation model (Howat et al., 2014) at intervals

of 0.05 degrees from 0 to 0.8 degrees (see Figure 2). An error model was fit following (Schröder et al., 2018; Smith et al., 2020) to obtain the overall noise level of each mission and their sensitivity to surface slope. The error model was taken to be:

$$\sigma_i = \sigma_{noise} + \sigma_{slope}\, \alpha^2 \qquad (2)$$

where $\sigma_i$ is the binned standard deviation, $\sigma_{noise}$ is the estimated noise level at zero slope, $\sigma_{slope}$ is the slope dependent error (sensitivity) and $\alpha$ is the surface slope in degrees. The error model was fit from 0 to 0.5-degree surface slope to obtain a good fit: bins with slopes > 0.5-degree have higher variability (Figure 2) due to lower data counts. Model fit results are summarized for each mission in Table 1.





**Table 1: Estimates of elevation error for different missions and modes through comparison with airborne measurements. Elevations**
**within 50 m of each other were selected with a maximum time difference of three months. Tabulated below is the estimated noise**
**level ($\sigma_i$) of each dataset and its corresponding error as a function of surface slope ($\sigma_{slope}$). The errors were generated by computing**
**the standard deviation of the difference as a function of surface slope at 0.05-degree intervals from 0-0.5 degrees.**

| Mission | $\sigma_{noise}$ (m) | $\sigma_{slope}$ (m deg$^{-2}$) | Count |
|---|---|---|---|
| ERS-1 (Ocean) | 0.24 | 21.32 | 678 |
| ERS-1 (Ice) | 0.37 | 30.21 | 6963 |
| ERS-2 (Ocean) | 0.06 | 19.91 | 540 |
| ERS-2 (Ice) | 0.41 | 29.85 | 35,734 |
| Envisat | 0.15 | 34.85 | 62,320 |
| ICESat | 0.11 | 0.71 | 7,192 |
| Cryosat-2 (LRM) | 0.12 | 23.70 | 25,587 |
| CryoSat-2 (SARIn) | 0.34 | 1.26 | 36,346 |
| ICESat-2 | 0.07 | 0.43 | 123,951 |

The estimated noise level is used as a measure of the systematic error of each mission or mode. The random error is estimated
from the variability of the elevation change data for each monthly estimate in the spatio-temporal stacking procedure (Nilsson
et al., 2022). To estimate the total error for each monthly estimate of elevation change the two error components can be
combined using root-sum-squares (RSS) after scaling the random error using an appropriate correlation length. The random
error is then used as input in the gridding procedure as the measurement error and added to the diagonal elements of the
covariance matrix. The final elevation change product includes monthly fields of both random and systematic error. This
allows the user to derive an independent error budget if needed (i.e., to account spatial auto-correlation in the random error).

To estimate the accuracy of long-term elevation change rates, and to maximizing spatial coverage, elevation change rates were
estimated from the ATM data over the time period 1993-2019, following the approach of (McMillan et al., 2016; Nilsson et
al., 2022; Wouters et al., 2015). Elevation change estimates were computed by fitting a linear model to the elevation data inside
a search radius of 175 m around each ATM reference track location. The data was centred to 2006 and the solution was
discarded if the estimated rate was larger than 20 m a$^{-1}$, if the number of points in the solution was less than ten, or if the
maximum time difference was less than five years. The derived rates where then averaged to the same spatial resolution as our
height change product (1920 m) and subsequently differenced with the altimetry (altimetry minus airborne). The results of the
elevation change rate validation are shown in Figure 3.






To estimate errors in rates of mass change for different time intervals we followed the approach of Nilsson et al. (2022). For each region of interest (drainage basin, peripheral or continental estimates) the mass change rate error was computed by taking the mean random and systematic elevation errors for each time series inside the specified time interval. The random error was then divided by the square root of the number of un-correlated spatial bins estimated as:


$$N = {A_{ROI}}/{\pi r^2} \tag{3}$$

where $A_{ROI}$ is the total area of the region of interest (ROI) and $r$ is the correlation length of the data. In this study we set $r = 40\ km$. The correlation lengths were selected from a covariance analysis of elevation changes for each monthly spatial field for the two regions. The systematic and random error are then added in quadrature and the corresponding mass rate ($\varepsilon_{rate}^2$)

error for each time interval is computed by dividing by the desired time span.

Estimating FAC errors is an active area of research and not within the scope of this study. Instead, we take the difference in FAC change between the two models at the measure of error. The error in the derived firn-rate for each time interval is then estimated as the RMSE (including bias and variance) of the volume difference of the FDM models divided by the time span. The variance in the RMSE computation was scaled by the number of un-correlated bins, as done previously, using a correlation

length of 200 km defined from a covariance analysis. To derive mass change errors in gigatons per year we added the volume change error and FAC error in quadrature then multiplied by the density of ice.



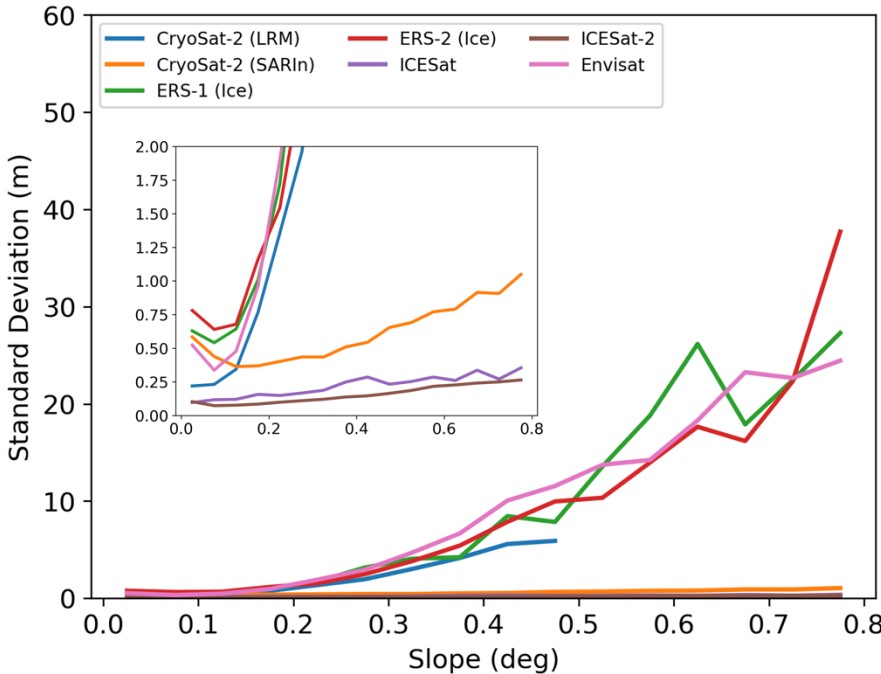

**Figure 2: Mission dependent elevation error as a function of surface slope. The figure highlights the recent improvements in ice sheet altimetry with the introduction of both laser (ICESat and ICESat-2) and interferometric synthetic aperture radar (CryoSat-2 SARIn mode). See Table 1 for derived parameters.**





**Figure 3: (Left) Difference in elevation change rates over the Greenland Ice Sheet between our synthesis height change product and ATM airborne laser altimetry covering the period 1993-2019 (panel a). (Right) Corresponding statistics and surface slope dependencies of the differences. ~51,000 points have been used in the analysis to generate long-term statistics covering a diverse range of slopes and surface conditions. An overall difference of 0.30 ± 13 cm a⁻¹ (panel b) is found over the entire region with an error of ~2.0 cm a⁻¹ for the flat regions increasing to ~40 cm a⁻¹ in the high-slope regions close to the margins of the ice sheet (c and d). Further, the effect of the background model is evident in the bias (panel c) where a decrease in bias can be observed in the high-slope areas of the ice sheet.**



## 4 Results

Our estimates of elevation change (Figure 1 and 4) are in line with previous studies (Khan et al., 2022; Otosaka et al., 2023)
for areas of large change, such as Jakobshavn, Helheim, Kangerlugssuaq and Storstrømmen glaciers. However, by
incorporating the background model, guided by velocity and hypsometry, we have achieved higher spatial resolution in regions
characterized by pronounced dynamic thinning, such as Jakobshavn, Helheim, and Kangerlussuaq glaciers. This is noteworthy
considering that many previous altimetry-derived studies have relied on more classical interpolation algorithms (e.g., kriging,
Inverse distance weighted (IDW) interpolation etc.), resulting in the spatial smoothing of these highly localized signals. The
applied background model addresses spatial sampling biases inherent to radar satellite altimeters, which tend to under-sample
troughs and valleys. By comparing the long-term trend differences between the velocity and elevation guided solution (i.e.,
using a background model) versus ordinary kriging/optimum interpolation, we find a difference here of -40 km$^3$ a$^{-1}$ with the
background model approach providing more negative results. Further, validation of the effectiveness of this technique is
demonstrated in Figure S1 of the supplementary material, where a comparison with ATM-derived rates was conducted.
Notably, the selected background model exhibited significantly reduced overall bias when analyzing steeper slopes (slopes >
0.5 degrees).

The accuracy, or noise levels, of the independent missions vary depending on their age. Older missions like the ERS exhibit
an overall ice sheet-wide accuracy of approximately 40 cm (Figure 2 and Table 1). In contrast, the more recent missions
demonstrate improved accuracy ranging from 7 to 15 cm, except for CryoSat-2 SARIn, which has an accuracy of around 35
cm due to its operational focus on areas with complex topography (Figure 2 and Table 1). Most notably, difference can be
observed between radar and laser derived surface elevations where laser outperforms radar over all slope intervals. It should
however be noted that CryoSat-2 SARIn mode provides significant improvements compared to traditional pulse-limited
altimetry. Over the slope range of 0.1-0.8 degrees CryoSat-2 show a higher slope dependency with an error ranging from 35
to 100 cm while ICESat and ICESat-2 show an error ranging from 10 to 20 cm. The impact of mission quality can also be seen
in timeseries error bars shown in Figure 5, most notably the reduction in error after the launch of ICESat-2 in 2018.

Analyzing the overall quality of our product, in the form of long-term elevation changes rates, we find a difference of 0.3 ±
12.8 mm a$^{-1}$ based on a comparison to ~51,000 ATM samples plus an additional slope dependent error of 1-38 cm a$^{-1}$ (Figure
3). Error is roughly 50% lower for our "background" interpolation algorithm compared to ordinary optimal interpolation.
These results vary regionally and, as observed in Figure 3, areas of rapid change such as the GrIS outlet glacier discrepancies
still exist. We also compare our product to PRODEM (Winstrup, 2023) over the 2019–2023 period. PRODEM consists of five
annual (summer data) gridded elevation maps posted at a 500 m resolution. The dataset covers the marginal zone of the
Greenland Ice Sheet, extending 50 km inland. Both products are based the same CryoSat-2 and ICESat-2 for this period. For
comparison, PRODEM was averaged from its native resolution to the JPL 1920 m grid. Elevation change rates were then





computed for both datasets over the 2019-2023 time period. We find a mean difference of 7.5 ± 92.7 cm a⁻¹ (JPL minus PROMDEM), excluding all rates exceeding ±5 m a⁻¹. Much of the difference can be attributed to methods used for interpolation and difference in temporal resolution. PRODEM uses spatial kriging while we use spatial kriging with the addition of an elevation and velocity dependent background model. Determining which dataset is more accurate is challenging, as we lack

external data, such as from ATM, to compare over this period.

To estimate mass change, we examine three regions of interest (ROIs): (1) the continental ice sheet, (2) peripheral glaciers and (3) individual drainage basins (Figure 1) by computing a linear trend for each ROI and time period. For the continental ice sheet, we find an overall loss of -173 ± 19 Gt a⁻¹ over the 1992-2022 time period. Losses are concentrated in the main outlet

glaciers in Greenland, such as the Jakobshavn, Helheim, N79 and Kangerlussuaq glaciers, and in coastal areas by the Denmark Strait on the Blosseville coast (Basin 3) and in the North-West between the city of Upernavik and Pituffik Space Base by Baffin Bay (Basin 8). These same regions have also experienced accelerations in rates of ice discharge except for the Jakobshavn Glacier which slowed due to recent ocean cooling (Khazendar et al., 2019), which can be seen in the Figure 4 panel 2017-2021. The first decade of the record (1992-2001) the ice sheet gained volume at rate of +51 ± 73 Gt a⁻¹. The ice

sheet then began rapidly losing volume in 2004, leading to an extreme rate of volume loss for the second decade of -321 ± 51 Gt a⁻¹. The mass loss was particularly large over the 2007-2011 time period (Figure 5). In the last decade of the record (2012-2022) there rate of loss slowed to -134± 39 Gt a⁻¹, a rate roughly 60% less negative than the proceeding decade (Figure 4-5) modulated by the slowdown of some outlet glaciers due to ocean cooling (Khazendar et al., (2019) and positive Surface Mass Balance (SMB) anomaly as seen in Figure S2 and S3.


Large difference in volume change rates can be observed at the basin scale. The general pattern depicts one of less mass loss in the North increasing southwards, with higher rates of loss in the West compared to the East. Basin 2 is the only region that experienced minimal change in volume over the three decades of observation (1 ± 4 Gt a⁻¹). All basins show a relative stable or positive rates of volume change in the first decade of the record. Nearly all Basins entered a regime of volume loss initiated

around 2004, except for Basin 6 where the onset of rapid volume loss was delayed until summer of 2009. The majority of eastern basins experienced a reduction in their rates of loss after 2012 (Basins 2-4). This is also true for the two most Southern Western Basins (5 and 6). Several years (1995, 2012 and 2019) experienced pronounced rates of volume loss instigated by warm summers that produced anomalously high rates of melt water runoff (see SMB anomaly in Figure S3).

The peripheral glaciers show an overall volume loss of -23 ± 5 Gt a⁻¹ for the period 1992-2022 (Figure 5), with annual rates modulated with changes in SMB (see Figure S3). Between 1992-2002 there was relatively little change in peripheral glacier volume, with losses beginning circa-2003. Greenland's periphercial glaciers have not been studies with the same intensity as the continental ice sheet due to challenges associated with sampling and the accuracy of retrievals (similar for altimetry, gravity and input-output methods). However, a few studies have focused on these glaciers over different time intervals; Khan et al.,



(2022) computed a total loss of -27 ± 6 Gt a[-1] from ICESat over the period 2003-2009 and -42 ± 6 Gt a[-1] for ICESat-2 over the period 2018-2021. This is almost 40% less than our estimate of -48 ± 19 Gt a[-1] over the ICESat period. However, over the ICESat-2 period our result of -37 ± 27 Gt a[-1] fits well within the 1σ error. The estimate from this study also aligns with the values obtained by Gardner et al., (2013) of -38 ± 7 Gt a[-1] over the period 2003-2009, but are larger than Bolch et al., (2013) of -28 ± 11 Gt a[-1] compared to our -40 ± 22 Gt a[-1] for the period Oct-2003 to March-2008. Further comparisons with estimates

generated from high-resolution DEM differencing yielded estimates in line with this study. Hugonnet et al., (2021) estimated a total mass loss of -36 ± 6 Gt a[-1] for Greenland's peripheral glaciers over the 2000-2019 period aligning well with our estimate of -29 ± 6 Gt a[-1]. Lastly a comparison with modelled results from peripheral glaciers (Schlegel et al., 2016) over the time period of 2003-2012 provided an estimate of -37 ± 25 Gt a[-1]. This aligns well with the estimate from this study of -43 ± 13 Gt a[-1] over the same period.


We further compare our result to those obtained from satellite gravimetry (GRACE and GRACE-FO). For the period 2002 to 2022 gravimetry yields an ice sheet plus peripheral glacier mass change rate of -276 ± 21 Gt a[-1] (NASA/JPL, 2019; Watkins et al., 2015). For this comparison the total extent of the periphery was used as GRACE missions can't distinguish between peripherical or continental glaciers. Combining the results of the peripheral glaciers (-25 ± 5 Gt a[-1]) and the continental ice

sheet (-250 ± 22 Gt a[-1]) over the same period yielded an estimate of -275 ± 23 Gt a[-1] aligning well with results from gravity alone.





**Figure 4: Greenland ice elevation change rates and acceleration from our synthesis of multi-mission altimetry for the 3 decades spanning 1992 to 2023. Decadal rates of elevation change are generally positive over the first decade (1992-2001), extremely negative in the second decade (2002-2011), and moderately negative in the third decade (2012-2021). A slowdown of Jakobshavn glacier, due to lower ocean temperatures at its front (Khazendar et al., 2019), can be seen for period 2017-2021 and in the map of acceleration (indicated on the maps with arrows).**




**Figure 5: Time series of glacier volume anomaly for the continental ice sheet, eight ice sheet drainage basins and peripheral glaciers for the period 1992-2023. Errors are shown with grey shading. Significant improvements in the errors can be seen with the introduction of ICESat-2 data starting in 2018.**

**Table 2: Rates of glacier mass change for Greenland Ice Sheet, its basins and peripheral glaciers over the period of 1992-2022 for various epoch.**

| Regions | 1992-2001 (Gt a$^{-1}$) | 2002-2011 (Gt a$^{-1}$) | 2012-2021 (Gt a$^{-1}$) | 1992-2022 (Gt a$^{-1}$) | 2002-2022 (Gt a$^{-1}$) | 2003-2009 (Gt a$^{-1}$) | 2000-2019 (Gt a$^{-1}$) | 2003-2012 (Gt a$^{-1}$) | Area (km$^2$) |
|---|---|---|---|---|---|---|---|---|---|
| **Basin 1** | -8 ± 13 | -13 ± 9 | -7 ± 7 | -10 ± 4 | -14 ± 4 | -5 ± 13 | -14 ± 5 | -19 ± 9 | 351,860 |
| **Basin 2** | 14 ± 14 | -8 ± 9 | 0 ± 8 | 1 ± 4 | -2 ± 4 | -6 ± 13 | -3 ± 5 | -6 ± 8 | 510,898 |
| **Basin 3** | -22 ± 18 | -90 ± 13 | -8 ± 10 | -38 ± 5 | -47 ± 6 | -98 ± 19 | -52 ± 7 | -97 ± 13 | 321,045 |
| **Basin 4** | -6 ± 18 | -41 ± 10 | -16 ± 8 | -21 ± 5 | -33 ± 4 | -51 ± 14 | -33 ± 5 | -50 ± 9 | 165,637 |
| **Basin 5** | 26 ± 9 | -22 ± 7 | -14 ± 4 | -13 ± 2 | -21 ± 3 | -15 ± 10 | -23 ± 3 | -22 ± 7 | 55,587 |
| **Basin 6** | 43 ± 13 | -36 ± 8 | -19 ± 6 | -19 ± 3 | -39 ± 4 | -14 ± 12 | -40 ± 4 | -44 ± 8 | 290,750 |
| **Basin 7** | 1 ± 15 | -47 ± 11 | -27 ± 6 | -30 ± 4 | -41 ± 4 | -35 ± 16 | -44 ± 5 | -46 ± 10 | 407,535 |
| **Basin 8** | 3 ± 21 | -60 ± 14 | -40 ± 9 | -40 ± 5 | -50 ± 6 | -61 ± 19 | -53 ± 7 | -65 ± 13 | 423,022 |
| **Periphery** | -5 ± 21 | -41 ± 13 | -10 ± 6 | -23 ± 5 | -25 ± 5 | -48 ± 19 | -29 ± 6 | -43 ± 13 | 87,836 |
| **Continental** | 51 ± 73 | -321 ± 51 | -134 ± 39 | -173 ± 19 | -250 ± 22 | -290 ± 71 | -264 ± 27 | -354 ± 48 | 2,526,423 |

**5 Discussion**

Our ice sheet altimetry synthesis methodology closely follows that of Nilsson et al. (2022) with a notable enhancement of a new background model. Use of a background model was previously integrated into Nilsson et al., (2022), but failed to yield improved results. In this iteration, significant effort was dedicated to refining background model. Various approaches were explored, and promising results emerged when the overall search radius was expanded to approximately 300 km, effectively accommodating velocity and hypsometric gradients within the data. This augmentation proved beneficial in capturing

correlated trends while preserving local behavior at smaller scales. Improvements to the background model help to better capture the elevation change dependance on glacier dynamics via velocity, and to more accurately capture elevation change dependence on elevation that is associated with surface mass balance gradients. The hypsometric component within the background model is of particular importance for peripheral glaciers, compensating for issues related to data quality and coverage over high-relief terrain. Use of elevation to guide the interpolation of elevation change data has been widely employed

in the study of glaciers (Gardner et al., 2011; Moholdt et al., 2010, 2012; Nuth et al., 2010) and can be credited with the close agreement observed in this study with several other studies investigating Greenland's peripheral glaciers ((Bolch et al., 2013; Gardner et al., 2013; Hugonnet et al., 2021; Khan et al., 2022; Schlegel et al., 2016)). We believe the difference in our estimated magnitude of mass loss over the 2003-2009 period, compared to the estimates Bolch et al. (2013; Gardner et al. (2013) and Khan et al. (2022), is attributed to improved data availability and coverage from the Envisat mission included in the study.

Although the data is of lower quality, it improves both the spatial and hypsometric coverage below the equilibrium-line altitude during this period, compared to ICESat alone. This is especially true in the southern parts of Greenland, where ICESat tracks



diverge and have less coverage. This improvement was supported by an area-hypsometry analysis, which showed more coverage for Envisat at lower elevations (<1000 m). The improvement in long term elevation change rates with the inclusion of velocity in the background model is evident in Figure S1, which illustrates a reduction in overall bias between the datasets

derived from our model and those from airborne laser altimetry. Improvements in the background model have enabled us to extend the record of peripheral glacier elevation changes. However, we posit that further refinement may be achievable by incorporating a more sophisticated version of our Gaussian process scheme. The empirical selection of the 150 km search radius or 3000 closet data points was aimed at capturing correlated gradients while also preserving local behaviors. Nonetheless, we maintain that further enhancements may be attainable through the incorporation of additional external data

and more robust gaussian process approach. It is important to emphasize that a background model that utilizes a "remove-and-restore" approach is not novel, as it has been widely employed in geodesy for decades (Moritz, 1978). Further needed enhancement would be the improvement of the slope correction algorithms currently used. This is especially important for the older pulse-limited mission, is still one of the largest error sources in altimetry, and has not been advanced upon in almost two decades.


One significant contribution of our new product is the inclusion of peripheral glaciers. To our knowledge, our product is the first to provide estimates of long-term elevation change of these glaciers at the spatial and temporal resolution provided here. The estimates derived from these products demonstrate strong alignment with findings from various studies conducted over different time spans (Bolch et al., 2013; Gardner et al., 2013; Hugonnet et al., 2021; Khan et al., 2022). Despite their relatively

small area (roughly 4% of Greenland's ice cover - Khan et al., 2022), peripheral glaciers are found to contribute roughly 12% of the total ice loss from Greenland over the last three decades. Their inclusion enables a more accurate comparison between altimetry-derived estimates of mass change and those obtained through gravity measurements. The current resolution of GRACE /-FO, on the order of 300 km, poses challenges to isolate peripheral glacier mass change from the ice sheet mass change (Otosaka et al., 2023). Consequently, GRACE /-FO is only able to measure the combined peripheral plus ice sheet

mass change. We find good agreement between our Greenland-wide estimate of glacier mass change with those derived from GRACE over the period 2002-2022. This alignment underscores the importance of integrating peripheral glaciers into assessments of Greenland's mass change.

Modeled FAC stands as one of the most critical components for accurately estimating the mass balance of ice sheets, yet it has

persistently remained one of the largest sources of error. Over the past few years, the scientific modeling community has made significant advancements in developing a variety of new models aimed at reducing this uncertainty. In this study, two such models have been employed to generate mass loss estimates and explore the relative uncertainty by quantifying model spread. Incorporating several of these models should provide a better understanding of the potential errors in model output. Therefore, this study adopts an agnostic approach, refraining from favoring one model over the other, under the belief that averaging

multiple estimates will yield a more accurate result. This limited comparison reveals that even employing just the ensemble



mean can yield satisfactory results. However, it is strongly recommended to conduct further analysis on how surface melt is incorporated into these models, given the different forcing and model physics employed in GEMB-FDM and GSFC-FDM as an example. This is evident in Figure S4, which highlights divergence around 2005 when surface melt becomes more prevalent.

The long-term record of Greenland glacier elevation change provided here is the most detailed record currently available. It depicts large variability in elevation change rates that are controlled by ocean-ice interaction and changes in SMB (Hanna et al., 2008; Khazendar et al., 2019; Wang et al., 2023). One interesting finding is the increase in mass of the GrIS in the first decade of the record. This mass increase is consistent with neutral to positive SMB anomalies after the 1995 melt event (Figure S3). Previous studies have either considered this time period stable or showing a negative mass balance (Mouginot et al., 2019;

Otosaka et al., 2023; Simonsen et al., 2021). This might be related to the temporal resolution these previous studies have used on the order of several years with large windows for aggregating data to reduce noise. Applying such large windows might have the impact of spreading 1995 melt event to nearby years or removing it depending on the method used. In Otosaka et al., (2023) the 1995 melt event can be seen in the input-output method in Figure-1 but not in the the final Figure-4 after generating the cumulative time series of ice sheet mass change by integration. Another interesting finding is that the well-documented

slowdown of the front of the Jakobshavn glacier (Khazendar et al., 2019) is well resolved in our product, providing strong evidence of the capability of this updated processing approach to capture these kinds of localized phenomena. Finally our long-term estimate for the continental ice sheet of -173 ± 19 Gt a$^{-1}$ fits well with the results from the latest "Ice sheet Mass Balance Inter-comparison Exercise" (IMBIE) of -169 ± 9 Gt a$^{-1}$ (Otosaka et al., 2023) which further indicates convergence of mass loss estimates for Greenland.

**6 Conclusion**

Our altimetry synthesis reveals a consistent pattern of surface elevation change rates across the Greenland Ice Sheet and peripheral glaciers, which aligns well with findings from previous studies. By integrating a background model that is guided by velocity and elevation, we manage to achieve higher spatial resolution, particularly in regions experiencing pronounced dynamic thinning. This advancement is crucial, as it allows for the preservation of small-scale and highly localized signals that were previously smoothed out by more traditional interpolation algorithms. Moreover, our background model effectively

addresses spatial sampling biases inherent to radar satellite altimeters, enhancing the accuracy of our derived elevation change rates. The accuracy of independent missions varies, with older missions exhibiting higher noise levels compared to more recent ones. Notably, significant differences exist between radar and laser-derived surface elevations, with laser altimetry outperforming radar across all slopes. While our interpolation algorithm demonstrates overall improved accuracy compared to

ordinary optimal interpolation, regional variations persist, particularly in areas of rapid change such as outlet glaciers. On a broader scale, our analysis of long-term volume changes reveals an overall loss of mass for both the continental ice sheet and peripheral glaciers. Specifically, we estimate a combined (ice sheet plus peripheral glaciers) mass loss of -196 ± 20 Gt a$^{-1}$ over

the period 1992-2022. These findings are consistent with previous studies and highlight the ongoing impact of climate change on Greenland's ice mass change. Furthermore, our results demonstrate the importance of accounting for changes in firn air
content when estimating mass balance, as the choice of firn model can significantly affect mass balance estimates. In summary, our study contributes to a deeper understanding of Greenland's response to a changing climate and underscores the critical role of satellite altimetry in monitoring long-term changes in ice mass. Moving forward, continued research and refinement of modelling techniques will be essential for further improving estimates of ice sheet and glacier mass change and their contribution to sea level change.


## 7 Author contributions

JN and ASG conceptualized the study. JN conducted the analysis, wrote the majority of the main text, and made all figures. JN and ASG contributed to conceptualization and algorithm development. All authors contributed to the writing and editing of the manuscript.


## 8 Code and Data availability

Data can be found in Nilsson and Gardner, (2024) (https://doi.org/10.5067/ICFVI7DKHZJV). The code and algorithm used to generate the product are part of the "Cryosphere Altimetry Processing Toolkit" (captoolkit) and can be found here https://github.com/nasa-jpl/captoolkit (last access: 2024-07-22).

## 10 Competing Interests

The contact author has declared that none of the authors has any competing interests.

## 10 Acknowledgements

The authors were supported by the ITS_LIVE project awarded through NASA MEaSUREs program, and the NASA Cryosphere program through participation in the ICESat-2 science team. We thank the NASA and the European Space Agency
(ESA) for distributing their radar altimetry data. The research escribed in this paper was carried out at the Jet Propulsion Laboratory, California Institute of Technology, under a contract with NASA. © 2024. California Institute of Technology. Government sponsorship acknowledged.

## 10 Financial supports

This research has been supported by the Jet Propulsion Laboratory (NASA MEaSUREs and NASA Cryosphere Science Program as well as the Jet Propulsion Laboratory, California Institute of Technology, through an agreement with the National Aeronautics and Space Administration.



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
