# Peer review of "Elevation Change of the Greenland Ice Sheet and its Peripheral Glaciers: 1992-2023"

_Earth System Science Data, 2024_

## Referee Comment (RC2)

**Review of Nilsson and Gardner: Elevation change of the Greenland ice sheet...**

An excellent and clearly written paper, with the a sound methodology of remove-restore. It is especially to see how useful the CryoSat SARin mode has performed relative to IceSat-2. This is a landmark paper on Greenland ice sheet changes.

It would be useful to mention the use of outlet glacier ice  velocity for elevation changes already in the abstract, since this is not a "clean" elevation change paper. Also, could be useful to include in one of the figures the outline of the peripheral glaciers - not all readers know the Bolsch et al reference, and the definition of "peripheral glaciers" there are controversial (a large part of the central East Greenland ice sheet south of Scoresbysund is e.g. classified as "glacier" even though it is not (just an area of ice sheet with Nunataks). Therefore the split quoted between "glaciers" and "ice sheets" is somewhat ambitious.

---

## Referee Comment (RC3)

Review:
Elevation Change of the Greenland Ice Sheet and its Peripheral Glaciers: 1992–2023

Johan Nilsson and Alex S. Gardner

The paper provides a new data set of elevation change of Greenland and its peripheral glaciers based on analysis of multi-mission satellite altimetry for the observational time period 1992 - 2023.

In addition, the authors provide numbers of ice mass loss using a firn compaction model to convert from volume to mass.
Main findings, next to a complete monthly time series of elevation change are very strong decadal changes in ice mass loss with a total loss of 6076 Gt within the period of 1992-2022.

The data set is accessible via the given link and the netcdf file is easy to read and fulfills the standards of ESSD.
The article supports the data set.
The data set itself is accessible, unique, complete, usable in its current format and useful for a wide community.
The paper is well written and structured, figures are of good quality. Some methodical explanations I did not fully understand. This is listed below.

In general, I would like to see the data set published as it provides a long term elevation change product for Greenland and peripheral glaciers, which is very important, but I still would like to see some improvements or clarifications – see below.

The authors put in a massive amount of work to provide this comprehensive dataset. Thanks to the authors.

However, I still have some questions and concerns which are listed below and hope that the authors can address the given points as I think that such a data set is of high interest for a wider community.

When opening the netcdf file using ncview all variables seem to be flipped in the y-direction. This is confusing, although the y-coordinates are in line with the flipped data fields. I think this should be improved.

[Figure]

As this is a companion paper to Nilsson (2022) most of the methods are already explained in detail. Improvements to the processing chain are addressed in this paper.

Improved input data products were used for CryoSat (Basline-E). For Envisat  ENVISAT GDR 3.0. was used including data after the ENVISAT orbit shift. The source of ERS1/2 is not mentioned. Are you using the REAPER product?
I have to mention, that the most updated data products for ERS1/2 and ENVISAT are not used in this processing. If an update is planned, please consider to use FDR4ALT reprocessed data. https://www.fdr4alt.org/ . At this stage I don't consider this as a major request, it's more a recommendation.

**In section 3.6** the radar penetration and the adjustment of seasonal amplitudes is explained. I still have some concerns if an artificial correction of seasonal amplitudes using the characteristics of another mission (here CryoSat is used as reference mission) is a good way to handle this problem. To my opinion, one should leave the data as they are after the correlation with backscatter, tail and LEW and not trying to reduce the seasonal amplitude by another mission. Especially the use of CryoSat-2 as reference I see critical, as the two modes (LRM and SARIn) are behaving different in terms of penetration. If such an additional amplitude suppression should be applied, then I would recommend to use ICESat2 instead. Please also include the reference to Helm et. al. 2024 who developed a new CNN based retracking approach to tackle the penetration issue directly within the retracking.

**In section 3.8** the interpolation using velocity and hypsometry is used and an update to 2022 paper is explained. I'm not really understand why such an 1/v^2 weighting is used and why you are not using the ALOG10(v) as in the 2022 paper. I thinks the risk is high to overestimate

elevation change when using the velocity directly as the gradients are very high.  I also have concerns, that a correlation which is found for velocities of 300 to 500 m/a can be used for other velocities. I personally think that the correlation is spatially highly variable and one should not transform a certain behavior which works in a specific region   to another region. But maybe I misunderstand something how you applied parameters of the muli parameter regression method.

I also don't understand why a biquadratic surface is fitted to the monthly elevation change for bin sizes of 100km. A biquadratic surface was already fitted to the point cloud to remove topography, which makes sense, but I don't see why an elevation change should behave like an biquadratic surface in such a large area. Is this fit only used for outlier rejection or are the fit parameters used for the interpolation?

Would it be possible to provide two correlation maps of Greenland to demonstrate the similarity of the linear behavior of hypsometry/Velocity against elevation change, to see the spatial variability? Why is a binning in 100m bands necessary and does it make sense in the interior of Greenland, or should a smaller bin size be used?

How does the background model looks like and does it change with time? As I understand, the background model is estimated for each bin at each monthly time step? If this is correct, then the background model is varying from month to month, which I think is questionable. Personally, I think that monthly anomalies are not correlated with hypsometry and velocity as those are more driven by changes in SMB and not dynamics.

So, I would recommend to remove a long term trend for each mission and only use this velocity/hypsometrie approach on elevation change trends as they represent a 'long term' dynamic behavior, which correlates with velocity and/or hypsometry.

The monthly anomalies or residuals should be interpolated by IDW, ordinary kriging or median interpolation and can then be added back to the interpolated trend. This would speed up the processing as well and minimize the risk of introducing high values in sparsely sampled areas at the margins for early missions, due to the high velocity gradient.

**Validation, section 4:**

Would it be possible to provide Altimetry – ATM differences of elevation change on decadal basis: 1992-2002, 2002-2012, 2012-2023. This would help to understand the large discrepancies to other studies especially for the ENVISAT/ICESat time period 2003-2012. Liang et. al. published a table and comparing volume changes derived by different authors.

**An elevation change dataset in Greenland ice sheet from 2003 to 2020 using satellite altimetry data**
Bojin Yang, Shuang Liang, Huabing Huang & Xinwu Li

https://doi.org/10.1080/20964471.2022.2116796

PUBLISHED ONLINE:
19 September 2022

Table 4 of 4
**Table 4. Comparison of total volume change rates for the GrIS derived by different studies.**

| Source | Data | Time span | dV/dt (km³/yr) |
|---|---|---|---|
| Felikson et al. (2017) | ICESat | 2003–2009 | −269 ± 37 |
| **This Study** | **ICESat** | **2003/02–2009/10** | **−210 ± 18** |
| Simonsen et al. (2017) | CryoSat-2 | 2010–2014 | −271 ± 32 |
| Smith et al. (2020) | ICESat&2 | 2003/10–2019/02 | −235 ± 3 |
| Li et al. (2022) | ICESat&2 | 2003/02–2019/09 | −192 ± 7 |
| **This Study** | **CryoSat-2** | **2010/08–2018/10** | **−335 ± 24** |
| **This Study** | **ICESat-2** | **2018/10–2020/12** | **−363 ± 20** |

In addition to this table Wouters (2008) and Velicogna (2009) published values derived from GRACE of 179 Gt/yr, 2003 Gt/yr respectively. Sörensen (2011), published values for the period 2003-2008 of -180 to -230 km3/yr. Ravinder (2024) publised values of -196km3/yr for the period 2011-2022.

I used your data set and estimated the volume change in the periods from 2003-2008, 2007-2011, 2011-2018, 2011-2022 and found for Greenland ice sheet a volume change of -337, -540, -170 and -165 km3/yr respectively.

Your data product exceeds literature values for the 2002-2011 time period by roughly a factor 2 but is showing slightly smaller values for the CryoSat2/ICESat2 time period. Especially the discrepancies in the ENVISAT/ICESat is in my opinion a major concern, which needs more careful evaluation. 540 km3/yr on average for 2007-2011 is even exceeding the extrem melt years 2012 and 2019. Where is this coming from?

In table 2 only the mass changes are given.
Can you please include a similar table with volume changes, as this would reflect the data product and also Figure 5 where volume changes are presented.

It would also be useful to add the volume change to each of the sub panels in Figure 4.

As you mention in the text and your results section mass change, it is necessary to provide exact information of how you derive mass change from volume change. How you apply the FAC and it would be useful to also provide the FAC data set in parallel with the elevation change product.

**In the results section** you explicitly highlight the background model approach using hypsometry and velocity. Hurkmanns (2012) already compared ordinary kriging with spatio-temporal kriging including external drift by using a velocity field to improve interpolation of sparse sampled area at Jacobshavn Isbrae. They found a 10-20% increase of volume loss and argued, that for the whole Greenland ice sheet less differences should be expected as not all outlet glaciers show such a high thinning rate and are much smaller. However, here we see an increase of up to >100% when compared with other studies. For the CryoSat2 / ICESat2 time periods where spatial sampling is much better at the margins most studies agree quite good with the values presented here. Therefore, I have some concerns about the results of the presented method in sparse sampled areas. The authors mention an orbit shift of ENVISAT 2010. Maybe this results in less coverage and therefore the method fails and provides unrealistic high values at the margins of the ice sheet. When looking at Figure 5, nearly in all Basins a very high loss can be observed between 2009 to 2011 and also between 2002 and 2004. The first period includes the orbit shift the second the combination of ENVISAT and ICESat. Maybe also the handling of velocity in the regression model not as log10 as you used for Antarctica is a parameter to look at.

Here I derived trends from your product and calculated the difference. Especially at the eastern margins where topography is complex and sampling in early missions is low you get extrem melt rates of sometimes 20 to 30m/yr. In the end this accumulates to those very high

ice volume loss, which I think is not correct. One can also see patterns of flipping colors, were some negative/positive values seem to dominate the interpolation.

[Figure]

Trend1: 2007-2011    Trend2: 2012-2023    Difference: Trend1 – Trend2

---

## Community Comment (CC1)

I'm curious what mask was used for both interior holes (nunatuks?) and the ice sheet boundary, and/or if it can be changed to address boundary issues.

Below are two images of the data product for essd 2024-311 and two images of the data product for essd 2024-348. This comment is submitted to both papers.

Interior holes can be filled (at possibly low quality) by interpolation. Missing data at the edges is harder to extrapolate.

This issue came up today after discussions with BedMachine about updates to that product. Updates need the mask and DEM to match temporally. The mask used here does not likely represent the best 'true' ice sheet mask – that would be its own product and needs to be evaluated separately. In fact none of the words 'mask', 'outline', and 'boundary' exist in 2024-348, while 2024-311 mentions using the Zwally (2012) ice sheet outline and RGI 7 for peripheral glaciers.

The Zwally mask does not capture the true edge in many places. Can this product be regenerated with a larger mask? Zwally but buffered by a few grid cells? The superset of Zwally, Mouginot, and BedMachine? What's the downside of having some land cells, or partial land cells, included in this, other than some locations where there should be no change?

If key grid cells are missing and cannot be reasonably interpolated or extrapolated, then this product cannot be used as a basis for an updated BedMachine DEM, and another product that does provide full coverage would need to be found and used instead.

There is an ongoing community effort to avoid under- or double-counting cells where RGI and ice sheet communities may overlap. Gaps in these products are counter productive to this effort.

Figures follow. Blue is BedMachine. Red outline is RGI. Orange is a 2022 remote-sensing 'best true' outline.

[Figure]

2024-311 Figure 2

2024-348 Fig 1

2024-348 Fig 2

---

## Author Comment (AC1)

**Elevation Change of the Greenland Ice Sheet and its Peripheral Glaciers: 1992–2023**

**Response to Reviewers**

We would like to thank the reviewers for taking the time to review our manuscript and apologize for the delay in our response. The delay was due to the reviews arriving at roughly the same time as my institutional transition, with my new affiliation not becoming active until September this year. In response to the reviewers' comments, we have made several updates to both the product and the manuscript. First, we updated the post-filtering to reduce issues such as noisy time series and cross-calibration errors. This has helped to mitigate some of the large month-to-month variations previously observed in the product. Second, we improved the combined masks for the periphery and continental ice sheet to ensure no overlap. Peripheral glaciers now only include weakly or non-connected glaciers (CL0 and CL1) from RGI 6.0. (There was a typo in the previous manuscript indicating that we used RGI 7.0.). Finally, we have included a digital elevation model in the product, combining altimetry-derived elevations with ArcticDEM data validated against ATM. This allows for studying changes in absolute topography. We have also updated the area in Table 3 for the continental ice sheet, as the previous value was incorrect.

Original text by reviewer:     *Black*

Response by author:     *Red*

**Response to Ken Mankoff**

*I'm curious what mask was used for both interior holes (nunatuks?) and the ice sheet boundary, and/or if it can be changed to address boundary issues.*

*Below are two images of the data product for essd 2024-311 and two images of the data product for essd 2024-348. This comment is submitted to both papers.*

*Interior holes can be filled (at possibly low quality) by interpolation. Missing data at the edges is harder to extrapolate.*

*This issue came up today after discussions with BedMachine about updates to that product. Updates need the mask and DEM to match temporally. The mask used here does not likely represent the best 'true' ice sheet mask – that would be its own product and needs to be evaluated separately. In fact none of the words 'mask', 'outline', and 'boundary' exist in 2024-348, while 2024-311 mentions using the Zwally (2012) ice sheet outline and RGI 7 for peripheral glaciers.*

*The Zwally mask does not capture the true edge in many places. Can this product be regenerated with a larger mask? Zwally but buffered by a few grid cells? The superset of Zwally, Mouginot, and BedMachine? What's the downside of having some land cells, or partial land cells, included in this, other than some locations where there should be no change?*

*If key grid cells are missing and cannot be reasonably interpolated or extrapolated, then this product cannot be used as a basis for an updated BedMachine DEM, and another product that does provide full coverage would need to be found and used instead.*

*There is an ongoing community effort to avoid under- or double-counting cells where RGI and ice sheet communities may overlap. Gaps in these products are counter productive to this effort.*

*Thank you for the question regarding the delineation of the ice sheet and its basins. To clarify, we used the Zwally basins solely for estimating drainage basin values; they were not employed to delineate the ice sheet itself. For the latter, we relied on the mask from Bolsh et al. (2013), as described in the manuscript. Regarding whether the data can be interpolated or extrapolated using new masks, we believe there is sufficient spatial and temporal coverage to perform such operations for example, for nunataks and the ice-sheet boundary, provided that the coverage of the "new" mask is not substantially different from the original. As touched upon, standardization of dataset resolution, reprojection, and re-gridding for this purpose is already being undertaken by the observational group within ISMIP7, and we anticipate that the outcomes of this project will facilitate intercomparison and broader use. We agree, however, that both time and effort should be devoted by the community to derive accurate and, ideally, time-varying masks. For future products, we plan to adopt updated delineations identified as community standards.*

**Response to Rene Forsberg**

*An excellent and clearly written paper, with a sound methodology of remove-restore. It is especially to see how useful the CryoSat SARin mode has performed relative to IceSat-2. This is a landmark paper on Greenland ice sheet changes.*

*It would be useful to mention the use of outlet glacier ice velocity for elevation changes already in the abstract, since this is not a "clean" elevation change paper. Also, could be useful to include in one of the figures the outline of the peripheral glaciers - not all readers know the Bolsch et al reference, and the definition of "peripheral glaciers" there are controversial (a large part of the central East Greenland ice sheet south of Scoresbysund is e.g. classified as "glacier" even though it is not (just an area of ice sheet with Nunataks). Therefore, the split quoted between "glaciers" and "ice sheets" is somewhat ambitious.*

*We thank the reviewer for the kind words regarding the study and the dataset, and we hope it will prove valuable to the community. We also appreciate the suggestion, and we have added text and a description related to the use of ice velocity and hypsometry to the abstract. Furthermore, we have included a figure in the supplement, including the new DEM, detailing the continental ice sheet and peripheral glaciers (see S6). While miniature versions of these regions are available in the time series figures, we elected to add a dedicated figure to clearly depict their coverage. We acknowledge that the definition of peripheral glaciers has been under discussion for some time. For consistency and ease of comparison with previous studies, we chose to use RGI-provided outlines for unconnected and weakly connected glaciers, as was done in the GLAMBIE project. This approach facilitates intercomparison between our results and those of other studies.*

**Response to William Colgan**

*Elevation vs. Mass Balance – The title and Figures 4 and 5 highlight elevation change through 2023, but the main abstract-level findings and Table 2 are mass balance through 2022. Perhaps during the review process, the FAC products will be updated by another year to allow the mass balance conversion to be updated through 2023 as well, but I wonder if it would be helpful for Figure 4 and 5 to show mass balance patterns and time series, rather than elevation patterns and time series? The ultimate goal, and discussion point, is mass balance, so it would be nice if this is visualized.*

*Luckily since the paper was submitted new versions of both firn models have been updated to the end of 2023 and has thus been included in the manuscript and analysis. Regarding updating the figures to show mass balance patterns we decided not to do this as the models provide very different results both in space and time, which why we also took the ensemble approach when generating mass balance estimate. We provide the elevation change patterns only to allow for better comparison with other products where signals are not masked or modified due to the model used. I have added an example of how the spatial pattern differin the supplement (S5)and there is also a figure detailing the temporal difference in the supplement (S4).*

*Altimetry Data – It might be helpful, either in the Introduction or Section 2, to give some characteristics of the different satellite missions to place them in context. For example, perhaps footprint size at a given elevation, along track shot spacing at a given elevation, temporal repeat, track spacing at a given latitude, etc. The lasers altimeters are really quite different from the radar altimeters in this regard, and perhaps it would be helpful for the reader to understand any differences between the radar altimeters?*

*Thank you for that suggestion. We have added a table (Table 1) that provides some key metrics providing the reader with a better overview.*

*Filtering – There a several filtering steps. Sections 3.5 and 3.7 both have sequential 10-sigma and +/- 10 m residual filters. I guess this probably removes only a small fraction of observations. But Sections 3.8 and 3.9 both refer to rejecting data beyond the inter-quantile range, meaning below 25th and above 75th. This seems to be happening sequentially. Does this mean removing 50% in 3.8 and then 50% again in 3.9, leaving only 25% of the original observations? In Table 1, you have a count of how many observations per mission are used in the cross over analysis. It would be super helpful to have a similar table showing how many observations you start with for each mission, and how many observations remain after filtering, or even the steps therein. It is hard to tell which filter is the main QC, but I guess it is 3.9.*

*Yes, there is, unfortunately, and fortunately, a considerable amount of filtering that was applied to obtain the current results. The filtering described in Sections 3.5 and 3.7 is performed on point data **before** generating the monthly mean estimates. In contrast, the filtering in Sections 3.8 is applied during the gridding process. This involves removing spatial residuals from a fitted surface, which eliminates only a small portion of the data, primarily large outliers, without affecting most observations. For 3.9 they were done in the time domain at each node after lowess trend had been removed, which again were aimed at removing gross-outliers. We followed this rule (Lower bound=Q1−1.5×IQR, Upper bound=Q3+1.5×IQR). Hope that answers the question. We have modified the description in*

*3.8 to reflect that. Unfortunately, the details of some of these filtering steps were not saved during processing as the effect was measured visually together with cal/val and mass balance comparisons.*

*Spatial Resolution – There is a stated "native output" of 1920 m, with several intermediary resolutions. I see the slope correction (Section 3.2) is being done at 2 km resolution, the topographic separation (Section 3.3) is being down at 500 m resolution, the scattering correction (Section 3.4) is done at 1 km. I am not sure what resolution is being used for the critical step of the regression merge (Equation 1; Section 3.8). This is presumably happening at the native output resolution, but this should be made explicit. How are the intermediary corrections and higher resolution regression datasets (velocity and DEM) are being interpolated? Is there an influence on the final product if intermediary corrections are done at better/differing spatial resolution?*

*We have added additional text in the beginning of the methodology and added text in each processing step (Section) to make it more clear of where its being applied and why. The velocity and DEM were averaged to the native 1920 m resolution as its needed for each grid node. Effect of binning at 100 m did help reduce variability for the velocity/hypsometry estimates and allowed for a better dh versus velocity/hypsometry relationship. But this was determined empirically. We have made changes to the manuscript to make these different steps clearer.*

*Correlation Lengths – Somewhat related to spatial resolution above, there are different correlation lengths adopted in different sections. For example, in the multi-mission stacking (Section 3.5) the correlation length is 2 km. With a 1.92 km resolution this is effectively one grid cell? In the multi-mission merging (Section 3.8) the correlation length is 20 km. Then in the post-filtering of the product (Section 3.9) there is reference to a correlation "filter length of nine points", perhaps meaning 9x1920 = 17.28 km? Then the error analysis (Equation 3) uses a correlation length of 40 km. Lastly, the FDM models are said to have a correlation length of 200 km, based on a covariance analysis. Perhaps the reader would appreciate just a couple sentences explaining why specific correlation lengths are being chosen? Or, put the other way, why is there not a uniform correlation length to represent the length-scale of change in this multi-step analysis?*

*This is a good question and as with spatial resolution we have added or modified the text in these sections. Hopefully this will clear up any confusion for the reader. Also thank you for catching "2 km" this should be 1920 m and was a typo from a previous product version.*

*Velocity and Hypsometry Regression – This approach of using higher resolution DEM and velocity to downscale altimetry observations seems super novel and promising! The reader would like to see the "clear relationship" between velocity and elevation change in the main methods here (L192). If this relation is linear, then why is it weighted as an inverse squared in the regression? The reader could also use a better description of the velocity and hypsometry data being used. Presumably they are each single products applied to the entire time period? Or is some attempt made to acknowledge that velocity and hypsometry have changed over the 30-year period? This also makes we wonder, if you are downscaling with velocity and DEMs of better than 500 m resolution, and topographic separation can be at 500 m resolution, why isn't the final product one of these better resolutions, instead of the rather curious 1920 m resolution?*

*This approach is based on kriging with a trend. In geodesy it is referred to as the remove-restore technique (pioneered by people in Denmark like Rene Forsberg). We have now added that in the supplementary material (S7). We have modified the text "clear relationship" to" relatively linear relationship" as its not fully linear, but more of combination of trend and exponential for many basins and time steps. But the range 1-300 m/yr usually shows a more linear behavior. This range is also less noisy than in the larger velocity (>300 m/yr) (dh more consistent), so we choose to put more weight in the lower range as the relationship is more stable. As we run this at each grid node a more complex non-linear model was not feasible due to the cost of computation. Hence, we chose to use a simple linear model weighted to the linear range. This is also more conservative avoiding runway estimates, as can be the case for exponentials. The velocity and elevation products are static products but usually based on multi-year data, but we make no assumption of change over time of these external data. We support this decision in the text with a reference to Hurkman et al. (2012). This background model is a form of downscaling, but we have chosen to keep the product at 1920 m, which is better representative of the native resolution of the older altimetry missions (1-3 km depending on mission – better for laser altimeters). In the future, we can absolutely increase the resolution if it is of value. The 1920 m resolution comes from the choice of the nested ITS_LIVE grid.*

*Novelty* – The novelty of these result are clearly perhaps their long temporal span and high spatial resolution, but the opportunity to actually highlight this is perhaps missed. For example, the Results begin (L304): "Our estimates of elevation change … are in line with previous studies … for areas of large change, such as Jakobshavn, Helheim, Kangerlugssuaq and Storstrømmen glaciers. However, by incorporating the background model, guided by velocity and hypsometry, we have achieved higher spatial resolution in regions characterized by pronounced dynamic thinning, such as Jakobshavn, Helheim, and Kangerlussuaq glaciers." This is confusing. Are the results the same or different for these big glaciers? Perhaps focus the reader on the new insight and higher spatial resolution in the pre-2003 period, specifically?*

*Thank you for this comment we have changed the text and provided a clearer explanation. It now reads: "However, by incorporating the background model, guided by velocity and hypsometry, we have achieved improved characterization of the magnitude of the pronounced dynamic thinning".*

*Simonsen2021 (https://doi.org/10.1029/2020GL091216)* – This previous study also pulled together Ku band ERS / Envisat and Cryosat radar altimetry observations to create a 1992-2020 altimetry record of the ice sheet. It used also a non-traditional methodology, albeit very different methodology than presented here. It might be helpful reader to the Greenland altimetry reader to place the current multi-mission assessment in the context of this past multi-mission assessment. For example, at least a simple compare and contrast of ice-sheet wide timeseries of mass loss since 1992. Perhaps probing differences, if there are any, at sector scale?*

*We added a comparison of the overall long-term rates between our product and those of Khan et al. 2025 (https://essd.copernicus.org/articles/17/3047/2025/). This is the only comparable product in both spatial and temporal resolution with ours that we know of.*

*Nilsson2022 (https://doi.org/10.5194/essd-14-3573-2022)* – I appreciate that this is a companion paper to a previous Antarctic assessment, but at present there are perhaps >35

*references to Nilsson2022. This limits the reader's ability to quickly understand this new Greenland dataset with a stand-alone read. I would encourage the authors to scrutinize how many of these references could possibly be removed by adding a sentence or two of clarification into the current article.*

*Thank you for this comment. This is also something we have discussed and tried to mitigate in previous iterations.*

**Response to Veit Helm**

*When opening the netcdf file using ncview all variables seem to be flipped in the y-direc+on. This is confusing, although the y-coordinates are in line with the flipped data fields.*

*I think this should be improved.*

*Thank you for finding this. We have tried to fix that accordingly and we hope that it should now work for different types of software.*

*Improved input data products were used for CryoSat (Basline-E). For Envisat ENVISAT GDR 3.0. was used including data afer the ENVISAT orbit shift. The source of ERS1/2 is not mentioned. Are you using the REAPER product? I have to mention, that the most updated data products for ERS1/2 and ENVISAT are not used in this processing. If an update is planned, please consider to use FDR4ALT reprocessed data http://www.fdr4alt.org/ . At this stage I don't consider this as a major request, it's more a recommendation.*

*Thank you for finding this oversight and we have added the source of this in the data section. We are aware of the FDR4ALT product and have begun to look at it for future updates of both the Antarctica and Greenland solutions.*

*In section 3.6 the radar penetrationon and the adjustment of seasonal amplitudes is explained. I still have some concerns if an artificial correction of seasonal amplitudes using the characteristics of another mission (here CryoSat is used as reference mission) is a good way to handle this problem. To my opinion, one should leave the data as they are after the correlationon with backscatter, TES and LEW and not trying to reduce the seasonal amplitude by another mission. Especially the use of CryoSat-2 as reference I see critical, as the two modes (LRM and SARIn) are behaving different in terms of penetration. If such an additional amplitude suppression should be applied, then I would recommend to use ICESat2 instead. Please also include the reference to Helm et. al. 2024 who developed a new CNN based retracking approach to tackle the penetration issue directly within the retracking.*

*We have added the Helm et al. (2024) reference to the section as requested. In response to the question regarding amplitude normalization—a very relevant point—we note the following: the correction is applied because the scattering correction alone is insufficient to equalize amplitude across missions. The differences in amplitude between missions are not physical; rather, they arise from differences in sampling characteristics and from mission-specific instrument effects (e.g., sensor design, instrument aging, drift). Without this normalization, we observe substantial differences in amplitude at both basin and ice-sheet scales, which directly impact the resulting estimates. CryoSat-2 was selected as the reference mission as it uses a radar altimeter with similar frequency and sampling characteristics, and it observes the same surface topography integrated over the footprint over kilometer scales.*

*However, given the that ICESat-2 now has collected over 7-years of data we plan to investigate its use as the reference for future products. Regarding differences between SARIn and LRM modes, we acknowledge that SARIn experiences slightly different penetration characteristics. However, most of the regions examined lie below the equilibrium line altitude, where such effects are less pronounced or more difficult to distinguish from topographic influences.*

*In Section 3.8, the interpolation using velocity and hypsometry is used and an update to the 2022 paper is explained. I do not really understand why such a $1/v^2$ weighting is used and why you are not using $\log_{10}(v)$ as in the 2022 paper. I think the risk is high to overestimate elevation change when using the velocity directly, as the gradients are very high. I also have concerns that a correlation found for velocities of 300–500 m/a can be applied to other velocities. I personally think that the correlation is spatially highly variable, and one should not transfer a certain behavior that works in a specific region to another region. But maybe I misunderstand something about how you applied the parameters of the multi-parameter regression method.*

*Our velocity and DEM regression is done spatially at each grid node making it spatially variable. We agree that using log(v) is appropriate when you have large range of predictor values, but in our case, we limit ourselves to 1-500 m/yr range, where we usually find a linear range between 1-300 m/yr. Restricting us to a linear model we find that we are quite conservative and avoid large overestimates. We have also restricted ourselves in the filtering to ensure we only have month to month difference of 50 meters to avoid any major overestimates. Examples of the relationship for velocity and hypsometry can be examined in Figure S7.*

*I also don't understand why a biquadratic surface is fitted to the monthly elevation change for bin sizes of 100 km. A biquadratic surface was already fitted to the point cloud to remove topography, which makes sense, but I don't see why an elevation change should behave like a biquadratic surface over such a large area. Is this fit only used for outlier rejection, or are the fit parameters used for the interpolation?*

*The fit is only used for outliers' rejection as we treat the dh (m) as point data. We have added an additional sentence to describe this in Section 3.8.*

*Would it be possible to provide two correlation maps of Greenland to illustrate the similarity of the linear relationship of hypsometry/velocity with elevation change, in order to see the spatial variability? Why is binning in 100-m elevation bands necessary, and does it make sense in the interior of Greenland, or should a smaller bin size be used?*

*We performed an ice-sheet-wide correlation-length analysis of elevation change over the full temporal range (1992–2023) and found a mean correlation length of approximately 40 km, as stated in the manuscript. Based on this result, we computed our background model using correlation lengths between 50 km and 150 km to identify the optimum by comparison with elevation changes derived from ATM where 150 km provided the lowest bias. We therefore assume that the true correlation length lies around this value. The choice of this larger search radius was also supported by our earlier work in Antarctica, where basin-wide estimates (regressions) were used to fill areas identified as poorer quality. Those results showed considerably better agreement with GRACE. Regarding the choice of 100-m*

*hypsometric bins, we determined empirically that this bin size yields the most robust results. For glacier studies, bin sizes of 50–100 m are typically used, depending on the noise characteristics of the data, and our findings are consistent with this practice.*

*What does the background model look like, and does it change with time? As I understand it, the background model is estimated for each bin at each monthly time step. If this is correct, then the background model is varying from month to month, which I think is questionable. Personally, I think that monthly anomalies are not correlated with hypsometry and velocity, as these are more strongly driven by changes in SMB and not dynamics.*

*It does change with time and location, and your description is correct. However, we do not fully agree with the statement that monthly anomalies do not correlate with hypsometry and velocity. Several studies have shown that this correlation does exist, and that both hypsometry and velocity can serve as proxies for SMB and dynamic processes. Hypsometric extrapolation is widely used in glacier studies, both for temporal analyses and for estimating dV/dt, and multiple studies have employed velocity relationships for interpolation (e.g., kriging with a trend). Those studies are cited in the manuscript in Section 5 (Discussion).*

*Therefore, I would recommend removing a long-term trend for each mission and using the velocity/hypsometry approach only on elevation-change trends, as they represent a long-term dynamic behavior that correlates with velocity and/or hypsometry. The monthly anomalies or residuals should then be interpolated by IDW, ordinary kriging, or median interpolation, and can subsequently be added back to the interpolated trend. This would also speed up the processing and minimize the risk of introducing artificially high values in sparsely sampled marginal areas for early missions due to high velocity gradients.*

*This is an interesting idea, and we will definitely look into it for a future version of the product. For this release, we think our monthly regression approach is sufficient, since doing the regression at monthly time steps already helps reduce the influence of long-term trends in the elevation-change signal. That said, your suggestion is very helpful, and we agree that separating long-term trends from short-term anomalies could bring additional benefits and is worth exploring further.*

**Validation, Section 4:** *Would it be possible to provide altimetry–ATM differences of elevation change on a decadal basis: 1992–2002, 2002–2012, and 2012–2023? This would help to understand the large discrepancies with other studies, especially for the ENVISAT/ICESat period (2003–2012).*

*Liang et al. published a table comparing volume changes derived by different authors. In addition to this table, Wouters (2008) and Velicogna (2009) published values derived from GRACE of 179 Gt/yr and 200 Gt/yr, respectively. Sørensen (2011) published values for the period 2003–2008 of −180 to −230 km³/yr. Ravinder (2024) published values of −196 km³/yr for the period 2011–2022.*

*I used your dataset and estimated the volume change for the periods 2003–2008, 2007–2011, 2011–2018, and 2011–2022, and found values for the Greenland Ice Sheet of −337, −540, −170, and −165 km³/yr, respectively.*

*Your data product exceeds literature values for the 2002–2011 time period by roughly a factor of two, but shows slightly smaller values for the CryoSat-2/ICESat-2 time period. The discrepancies during the ENVISAT/ICESat period are, in my opinion, a major concern and require more careful evaluation. A value of −540 km³/yr on average for 2007–2011 even exceeds the extreme melt years of 2012 and 2019. Where is this coming from?*

*In Table 2, only the mass changes are given.*
*Could you please include a similar table with volume changes, as this would better reflect the data product, as well as Figure 5 where volume changes are presented?*

*It would also be useful to add the volume change to each of the subpanels in Figure 4.*

*As you mention mass change in the text and results section, it is necessary to provide exact information on how you derive mass change from volume change, how you apply the FAC, and ideally to provide the FAC dataset in parallel with the elevation-change product.*

*Given the review and some of our own checks, we have provided a slightly updated dataset with additional post-filtering. We believe these updated numbers agree more closely with previous studies, although they remain larger in magnitude in some cases. Our results for 2003–2009 of −**320 km³/yr** align better with previous studies - Felikson et al. (2017) reported −**269 km³/yr**, and Csatho et al. (2014) reported −**277 km³/yr**. The overall range of values reported in the provided table is also quite large: ranging from **210-277 km³/yr** and depends mostly on differences in processing and interpolation. It is also important to note that this time period includes Envisat in our study, which provides improved hypsometric coverage in combination with the background model. Because of this improved coverage, we would expect higher magnitudes, as more of the mass loss is being resolved compared to ICESat only studies Our estimate for 2011–2014 of −**305 km³/yr** fits well with previous studies by Nilsson et al. (2016) (−**289 km³/yr**) and Simonsen et al. (2017) (−**271 km³/yr**). We acknowledge that some of our values fall outside the range reported in the literature, particularly during the Envisat/CryoSat-2 transition period of 2007–2011. Although this is a relatively short interval for estimating rates, we did test the method by turning off the background model (separate product without background model applied), and the spatial pattern and relative magnitude remained for these periods. We show similar spatial patterns for the 2007-2011 period when comparing to annual estimates from the same period from Khan et al. (2025) and Csatho et al. (2014). Our overall mass estimate for the 2003-2023 period aligns well with Khan et al. (2025) as well.*

*The choice of 2007–2011 is affected not only by the mission transition but also by the substantial changes in mass loss that occurred during 2003–2011. From GRACE, an estimate of −**331 Gt/yr** is found for 2007–2011 compared to our −**440 Gt/yr**. However, slightly shifting the estimates to 2009–2011 yields −443 Gt/yr[1] from GRACE and −438 Gt/yr from our analysis (including peripheral glaciers). This indicates that the Envisat/CryoSat-2 boundary is not responsible; instead, the large magnitude lies in the 2007–2009 period, where ICESat and Envisat overlap. That said, we agree that this period requires further future investigation. We have now provided tables for volume and firn changes in the supplementary material.*

***In the results section*** *you explicitly highlight the background-model approach using hypsometry and velocity. Hurkmans (2012) already compared ordinary kriging with spatio-temporal kriging that included an external drift using a velocity field to improve interpolation of sparsely sampled areas at Jakobshavn Isbræ. They found a 10–20% increase in volume loss and argued that, for the entire Greenland Ice Sheet, much smaller differences should be expected, since not all outlet glaciers exhibit such high thinning rates and most are considerably smaller. However, in the present study we see increases of more than 100% when compared with other studies.*

*For the CryoSat-2/ICESat-2 time periods, where spatial sampling at the margins is much better, most studies agree quite well with the values presented here. Therefore, I have concerns about the performance of the method in sparsely sampled areas. The authors mention an orbit shift of ENVISAT in 2010. This may have resulted in reduced coverage, causing the method to fail and produce unrealistically high thinning rates at the ice-sheet margins.*

*When looking at Figure 5, nearly all basins show very high losses between 2009–2011 and also between 2002–2004. The first period coincides with the orbit shift, and the second corresponds to the combination of ENVISAT and ICESat. It may also be worth examining the handling of velocity in the regression model, specifically the use of velocity directly rather than $\log_{10}(v)$, which you applied in Antarctica.*

*I derived trends from your product and compared the differences. Especially along the eastern margins—where the topography is complex and sampling by early missions is sparse—the product produces extreme melt rates, sometimes 20–30 m/yr. These values accumulate into the very large ice-volume losses reported, which I believe are unrealistic. One can also see patterns of alternating positive/negative interpolation artifacts ("flipping colors"), suggesting that outliers or sparse sampling may strongly influence the interpolation.*

*We have updated the post-filtering of the data to reduce the impact of potential cross-calibration, slope errors and other sources of noise. This has reduced the magnitude of the rates you highlighted, particularly in regions sensitive to data quality issues such as slope corrections. As a result, the updated dataset shows better agreement with previous studies than the earlier version. It is important to note that the "low" spatial coverage is exactly what our method is designed to address. We agree with Hurkman et al (2012) that velocity should only help improve estimates in highly dynamic areas. This is also what we see in Figure 3f where the discrepancy in bias is in areas of high slope with a lower number of samples (in this case defined by the number of ATM comparisons). Most of the actual improvements come from the hypsometry as this is the main driver to overcome poor spatial coverage, which is quite stable and capped at zero elevation (see Figure S7) and thus very conservative. Comparisons with previous methods or studies, which were limited by sparse coverage, may naturally yield lower estimates in this case, especially for the Envisat/ICESat era. The 2002–2004 period, as well as 2009–2011, represent times of substantial mass loss. During these intervals we used both ICESat and Envisat (2002–2012), which improves hypsometric coverage by providing more data across all elevations. In combination with the velocity/hypsometry model, this can result in higher mass-loss estimates. Short periods, such*

as 2–3 years, are inherently noisy, whereas previous studies likely relied on smoother fields or time variable trends estimated from long-term polynomial fits. For the 2003–2009 period, our results show agreement with other studies. We acknowledge that future work should focus on improving fundamental aspects of the dataset, such as the slope correction and time series calibration, which is currently underway.